# Use of Pineapple Waste as Fuel in Microbial Fuel Cell for the Generation of Bioelectricity

**DOI:** 10.3390/molecules27217389

**Published:** 2022-10-31

**Authors:** Segundo Rojas-Flores, Renny Nazario-Naveda, Santiago M. Benites, Moisés Gallozzo-Cardenas, Daniel Delfín-Narciso, Félix Díaz

**Affiliations:** 1Escuela de Ingeniería Mecánica Eléctrica, Universidad Señor de Sipán, Chiclayo 14000, Peru; 2Vicerrectorado de Investigación, Universidad Autónoma del Perú, Lima 15842, Peru; 3Universidad Tecnológica del Perú, Trujillo 13011, Peru; 4Grupo de Investigación en Ciencias Aplicadas y Nuevas Tecnologías, Universidad Privada del Norte, Trujillo 13007, Peru; 5Escuela Académica Profesional de Medicina Humana, Universidad Norbert Wiener, Lima 15046, Peru

**Keywords:** microbial fuel cell, waste, pineapple, bioelectricity, *Wickerhamomyces anomalus*

## Abstract

The excessive use of fossil sources for the generation of electrical energy and the increase in different organic wastes have caused great damage to the environment; these problems have promoted new ways of generating electricity in an eco-friendly manner using organic waste. In this sense, this research uses single-chamber microbial fuel cells with zinc and copper as electrodes and pineapple waste as fuel (substrate). Current and voltage peaks of 4.95667 ± 0.54775 mA and 0.99 ± 0.03 V were generated on days 16 and 20, respectively, with the substrate operating at an acid pH of 5.21 ± 0.18 and an electrical conductivity of 145.16 ± 9.86 mS/cm at two degrees Brix. Thus, it was also found that the internal resistance of the cells was 865.845 ± 4.726 Ω, and a maximum power density of 513.99 ± 6.54 mW/m^2^ was generated at a current density of 6.123 A/m^2^, and the final FTIR spectrum showed a clear decrease in the initial transmittance peaks. Finally, from the biofilm formed on the anodic electrode, it was possible to molecularly identify the yeast *Wickerhamomyces anomalus* with 99.82% accuracy. In this way, this research provides a method that companies exporting and importing this fruit may use to generate electrical energy from its waste.

## 1. Introduction

The enormous technological advances of humanity have made electrical energy a necessity in carrying out our daily tasks [1]. Currently, the main sources of fuel for electricity generation are fossil sources (natural gas, oil, and coal), which are the cause of many health and environmental problems and affect the quality of life of many people. Even so, in 2019, fossil fuels comprised 85.5% of the energy produced globally, with India, Japan, China, and the US using 92.5, 90.8, 87, and 85.3%, respectively, being the countries with the highest dependence [2,3,4]. In this sense, the International Energy Agency (IEA) reported that in the year 2020, the consumption of electrical energy increased by 3% because of the pandemic (COVID-19), and that this percentage would increase by approximately 22% by the year 2025 [5]. Due to the increase in demand and the problems that it generates, several research groups have worked on different ways of generating electrical energy in an environmentally friendly way, including reusing different forms of waste and applying a circular economy in the process [6,7].

Microbial fuel cells (MFCs) are electronic devices that have become more relevant in recent decades, mainly because they use different types of waste to generate electricity, thus reusing a wide variety of biomass produced from the different activities of humans [8,9,10]. This type of cell consists of anodic and cathodic chambers, which are almost always connected inside by a proton exchange membrane and outside by an external circuit [11]. This system converts chemical energy into electrical energy through the oxidation and reduction processes in the anodic and cathodic chambers, respectively [12,13,14]. A great variety of substrates have been used as fuel in MFCs for the generation of electricity, while, on the other hand, due to the problems generated in the whole process of harvesting and consuming agricultural products, these are becoming a potential source for use as fuel in MFCs as there are a large number of microorganisms present in the generated waste that can produce bioelectricity [15,16,17]. It is estimated that 140 Gtons of agricultural waste is produced each year. For example, from just cereals, in 2019, it was estimated that 2.8 Gtons of waste was generated worldwide; for this reason, it is necessary to use technologies to reuse this type of waste [18,19].

One of the most consumed agricultural products worldwide is the pineapple (*ananas comosus*), which generated nine billion dollars worldwide in 2019 [20]. This fruit is generated in large quantities in South America, making this continent the main exporter worldwide; for example, in Brazil alone, 28,179,348 tons were harvested in 2019 according to the Food and Agriculture Organization (FAO) [21,22]. The increase in pineapple consumption is due to the high nutritional content of water, carbohydrates, organic acids, dietary fibers, antioxidants, vitamins, and minerals, as well as minerals such as calcium (Ca), magnesium (Mg), phosphorus (P), sodium (Na), manganese (Mn), iron (Fe), copper (Cu), zinc (Zn), and selenium (Se), and vitamins such as B1, B2, B3, B5, B6, B9, and C [23,24,25]. Research has been reported in which bioelectricity is generated through different types of organic waste; for example, banana and orange peel have been used as fuel to generate bioelectricity in MFCs with zinc and graphite electrodes, managing to generate peaks of 0.586 and 0.492 V in the cells with banana and orange substrates, respectively [26]. Similarly, Manjrekar et al. (2018), in their research, used kitchen waste in their double-chamber MFCs with aluminum electrodes, managing to generate voltage peaks of 365 mV. It was also observed that these values decreased over time, mainly due to the sedimentation of the organic components [27]. Likewise, tomato waste has been used as a substrate for the generation of electricity in single-chamber MFCs with zinc and copper electrodes, with voltage peaks close to 10.8 V and an internal resistance of 0.148541 ± 0.012361 KΩ being observed at the optimal acid pH for the operation of MFCs; this study is one of the most promising works to date [28]. It was observed that, in the investigations carried out, metallic electrodes are of great help in generating higher current and voltage values, while single-chamber MFCs obtain higher current and power density values [29,30]. In the reviewed literature, research that uses different types of fruit waste as fuel was found; however, the information on pineapple waste is very limited, although this material is very promising for the generation of bioelectricity in microbial fuel cells due to its physical, chemical, and biological characteristics. Therefore, it is important to expand the knowledge in this field through research using this type of waste.

The main objective of this research was to generate bioelectricity through single-chamber microbial fuel cells at the laboratory scale, using pineapple waste as fuel, and zinc and copper electrodes as anode and cathode electrodes, respectively. The microbial fuel cells were monitored for voltage, current, pH, electrical conductivity, and Brix degrees for a period of 32 days. We also found the values for power density, current density, and internal resistance of microbial fuel cells. Additionally, the FTIR (Fourier transform infrared spectroscopy) patterns of the initial and final substrate were studied, as well as the molecular biology of the biofilm formed on the anode electrode.

## 2. Materials and Methods

### 2.1. Manufacture of Microbial Fuel Cells

Three (03) MFCs were manufactured with acrylic tubes (Poly/methyl 2-methypropenoate)/PMMA) 10 × 30 cm in diameter and length, respectively. For the anodic and cathodic electrodes, copper (Cu) and zinc (Zn) 10 and 0.2 cm in diameter and thickness, respectively, were used. The Zn electrode was placed at one end of the tube with one side exposed to the environment and the other was exposed to the substrate used; while the anode electrode (Cu) was placed inside the MFCs, both electrodes were joined by an external circuit of Cu wire (0.25 cm in diameter) and a 100 Ω resistor, whose configuration was similar to that used in the work of Segundo et al. (2022) [28].

### 2.2. Pineapple Waste Collection

Pineapple waste was selected by the people who sell it at Mercado La Hermelinda, Trujillo, Peru, who managed to collect six kilograms in total. The collected waste was taken to the laboratories of the university in airtight bags and washed 3 times with distilled water to remove any type of impurities (e.g., dirt obtained from the market) acquired from the environment, and then left to dry in an oven at 30 ± 2 °C for 12 h. The pineapple waste was passed through an extractor (Labtron, LDO-B10- Camberley, UK) to obtain juice from the waste. It was possible to obtain 2 L of juice, which was placed in a beaker and stored until use.

### 2.3. Characterization of Microbial Fuel Cells

The voltage and current values were monitored using a multimeter (Prasek Premium PR-85) for 32 days, which has an external circuit with 100 Ω resistance. Power density (PD) and current density (DC) values were obtained using external resistors of 0.3 (±0.1), 0.6 (±0.18), 1(±0.3), 1.5(±0.31), 3(±0.6), 10 (±1.3), 20 (±6.5), 50(±8.7), 60(±8.2), 100(±9.3), 120 (±9.8), 220 (±13), 240 (±15.6), 330 (±20.3), 390 (±24.5), 460 (±23.1), 531 (±26.8), 700 (±40.5), and 1000 (±50.6) Ω, using the method of Segundo et al. (2022) [31]. The monitored values of electrical conductivity (conductivity meter CD-4301), pH (pH meter 110Series Oakton), and degrees Brix (RHB-32 brix refractometer) were also measured for 32 days. The transmittance values were measured by FTIR (Thermo Scientific IS50), and for the resistance values of the MFCs, an energy sensor (Vernier- ± 30 V and ±1000 mA) was used.

### 2.4. Isolation of Microorganisms from the Anode Chamber

To isolate the microorganisms, a swab of the anode plates was taken and then planted in McConkey agar and nutrient agar medium to isolate bacteria; on the other hand, Sabouraud agar was used to isolate fungi and yeasts with 4% glucose. Media were incubated for 24 h at 35 °C for bacterial isolations and 24 h at 30 °C for fungal and yeast isolations, and the procedure was performed in duplicate [32]. The reading consisted of observing the macroscopic characteristics of the colonies grown in the culture media, while methylene blue was used to observe the microscopic characteristics. Finally, pure cultures were made.

### 2.5. Molecular Identification of Fungi

Molecular identification was carried out in the BIODES laboratory (Laboratorio de Soluciones Integrales Comercial de Sociedad Limitada) where molecular biology techniques were used. The first users were the ITS (Internal Transcribed Spacer, USA) sequences, which are specific to fungi [33]. 

### 2.6. Statistical Analysis

The data points obtained from Figure 1, Figure 2 and Figure 3 represent the average values obtained from the three replicates, and the error bars represent the corresponding standard deviations. Meanwhile, the bioinformatic software MEGA X (Molecular Evolutionary Genetics Analysis, USA) analyzed the sequence obtained by comparing it with the sequences of reference yeast species. For this, the sequence alignment tool BLAST (Basic Local Alignment Search Tool) was used to identify the species based on the percentage of identity [32].

## 3. Results and Analysis

The monitored voltage values are shown in Figure 1a, where it can be seen that the voltage values increased from the first day (0.5685 ± 0.01 V) to day 16, when the maximum voltage peak was found (0.99 ± 0.03 V), and later decayed to 0.624 ± 0.03 V on the last day of monitoring. The high voltage values found are directly attributed to the microbiota found on the surface of the anode electrode, whose influence was described by Khan et al. (2017) [34]. Similarly, Waheed et al. (2016) showed that the size of the particles influences the rate of hydrolysis, limiting the generation of voltage [35]. This research found values that exceeded those obtained by Kalagbor et al. (2020), who, with the use of the pineapple substrate, generated 0.8 V during the first days; however, in said study, a tendency towards a decrease in voltage production was observed [36]. Likewise, Priya and Setty (2019) generated a peak voltage of 0.4 V on the seventh day from the use of apple juice as a substrate in the anode chamber. It is worth mentioning that the values obtained in this study are not higher than those found in this investigation [37]. Figure 1b shows the electrical current values monitored throughout the investigation, observing that the values increased from 0.09667 ± 0.00577 mA on day 1 to the maximum peak of 4.95667 ± 0.54775 mA on day 20 and then decreased until the last day (1.97333± 0.50213 mA). The values of electrical currents in the MFCs are governed mainly by the fermentative microorganisms that convert the substrate (fermented fuel), such as glucose, into small-chain organic acids, hydrogen, and carbon dioxide; electricity is generated at the same time as an interaction is formed between the reduced compounds that are produced under redox conditions during fermentation or possibly in some direct transfer of electrons between the microorganisms and the anode surface [38,39,40]. On the other hand, the values of the electric current in Figure 1b showed a decrease during the last days, which would be due to the diffusion of oxygen from the cathode to the anode due to the lack of a membrane between them [41].

**Figure 1 molecules-27-07389-f001:**
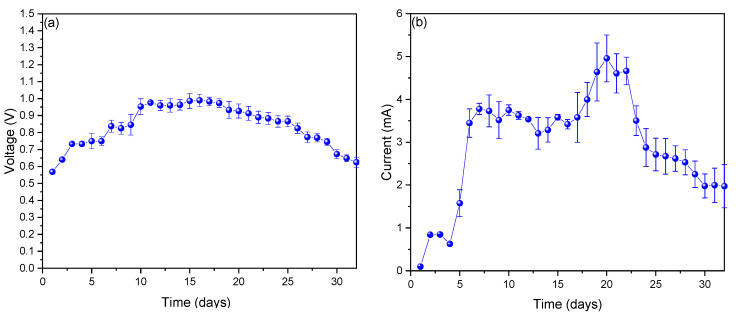
Monitoring of (**a**) voltage and (**b**) current values of microbial fuel cells (MFCs).

Figure 2a shows the values of the electrical conductivity of the substrates of the MFCs, observing that the values increased from the first day (189.3 ± 2.08 mS/cm) to the sixth day (199.33 ± 4.041 mS/cm) and then slowly decayed until the last day (135.51 ± 6.87 mS/cm). In previous studies, it was observed that the electrical conductivity values of different types of substrates (organic waste), when fermented, increase over time and decrease when observing substrate sedimentation (agglomeration of particles in the fermentation process) [42,43]. Figure 2b shows the pH values observed during the monitoring, where it is shown that throughout the monitoring, an acidic pH was maintained, although between days 16 and 19, maximum pH values were observed. For this work, it was observed that the optimal operating pH of the MFCs was 5.21 ± 0.18, which was achieved on day 16. In the literature, it has been found that the pH of the substrate affects the electrical efficiency of the MFCs because the transfer of electrons and protons occurs within them [44]. Thus, in the anode reactions, electrons are produced in the oxidation process, generating acidification, which consequently lowers the pH [45,46], while the reduction reactions that occur in the cathode produce alkalinization, increasing the pH, and the variation in the pH values over time is attributed to this [47,48]. In this sense, it is worth mentioning that high pH inhibits the growth of methanogens that indirectly improve the performance of MFCs [49]. The observed values of degrees Brix (°Brix) are shown in Figure 2c, which remained constant until the third day (6° Brix) and then gradually decreased until the twenty-fourth day, when they decreased to zero, and they remained constant until the last day. From monitoring, it has been observed in the literature that the °Brix values decrease mainly due to the decomposition of the nutrients of the substrates in the bioelectricity generation process of the MFCs [49,50].

**Figure 2 molecules-27-07389-f002:**
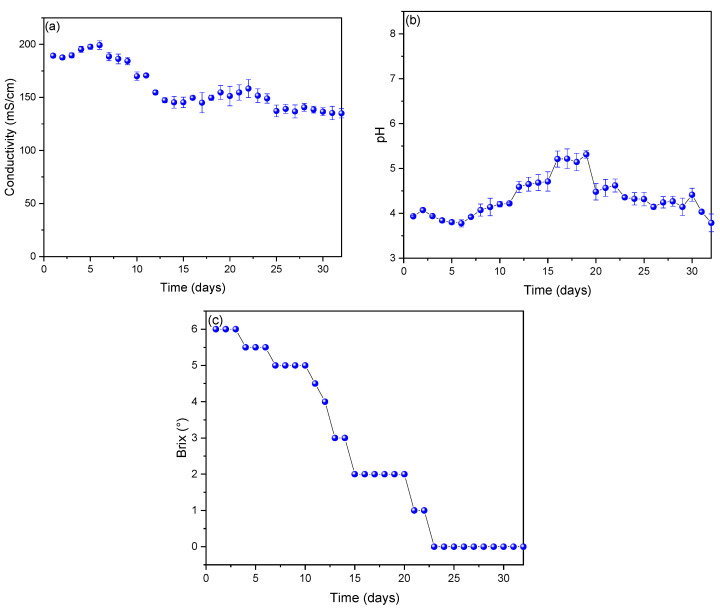
Monitoring of (**a**) conductivity, (**b**) pH, and (**c**) °Brix parameters of MFCs.

Figure 3a shows the value of the internal resistance of the microbial fuel cells, which are governed by Ohm’s Law (V = RI). The X axis represents the electrical current values (I) and the Y axis the voltage values (V), whose the linear fit slope would represent the average internal resistance (R_int._) of the MFCs. The R_int._ found was 865,845 ± 47.23 Ω in the MFCs; these values were found on day 20 because it was the day that generated the most intense electrical current values. Compared with other investigations, the values found for the R_int._ are high, but the values of current and voltage are higher than those found by other investigations; for example, Rashid et al. (2021), in their research, managed to generate 330 mV and 2.75 mA using single-chamber MFCs with graphite electrodes and pharmaceutical effluents with 99 Ω internal resistance [51]. Likewise, Liu et al. (2020) investigated the use of electrogenic bacteria as substrates and carbon electrodes in MFCs, managing to generate peaks of 0.63 V in the internal resistance of 162.9 ± 3.5 Ω [52]. One of the most important and influential factors that can explain this phenomenon is the presence of organisms that form the biofilm, mainly in the anodic electrode, since it is in charge of receiving the electrons [53,54]. Figure 3b shows the values of the power density (PD) and voltage as a function of the current density (CD), where the DP_MAX_ can be observed to be 513.99 ± 6.54 mW/m^2^ in a CD of 6.123 A/m^2^, with a maximum voltage of 874.46 ± 19.64 mV. These obtained values are high compared to those obtained by Xin et al. (2018), who managed to obtain 0.20 W/m^2^ in a CD of 0.27 A/cm^2^ and a peak voltage of 0.58 V [55]. The values obtained according to Huang et al. (2021) would be due to the high content of glucose as a carbon source for the microorganisms present in the substrates [56]. Additionally, the DP value found by Yaqoob et al. (2022) was 0.30 mW/cm^2^ at a DC of approximately 28 mA/cm^2^ in their dual-chamber MFCs using mango (Mangifera indica) debris as the substrate. All these values are lower than those found in this research, which highlights the importance of research and the use of pineapple waste and the electrodes used [56].

**Figure 3 molecules-27-07389-f003:**
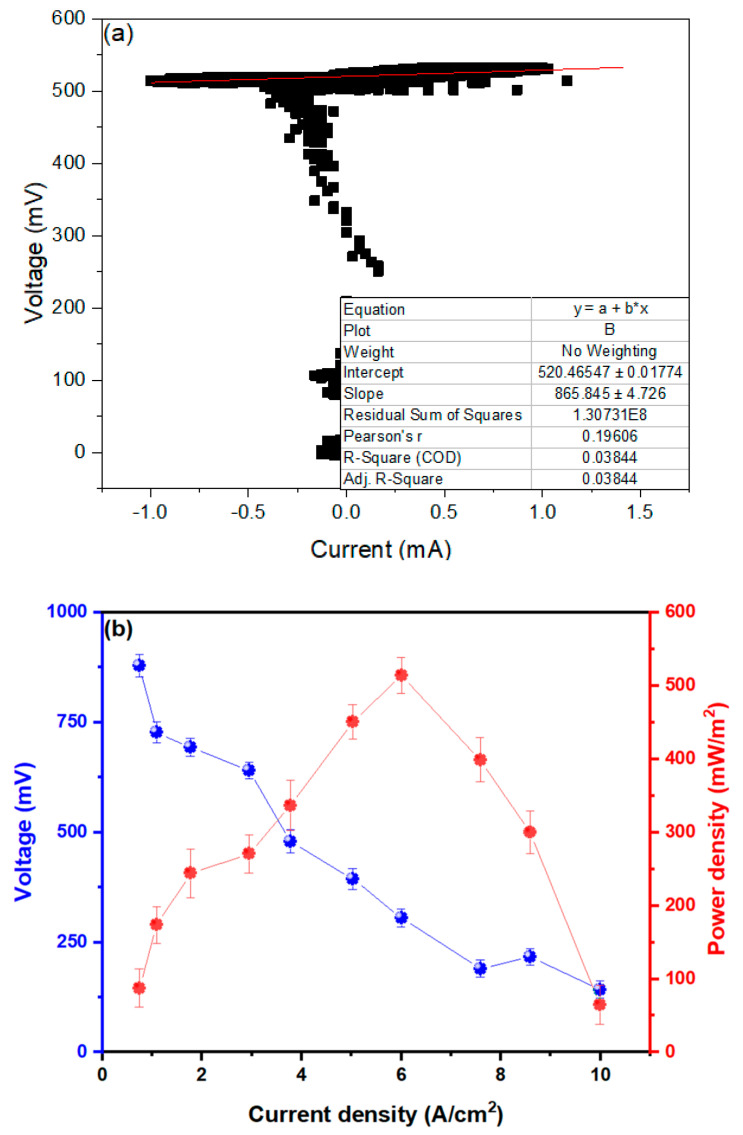
Characterization of (**a**) internal resistance and (**b**) power and voltage density in relation to current density of MFCs.

Figure 4 shows the transmittance spectra obtained by FTIR of the substrate used in the initial and final state of the monitoring performed. It was possible to observe that the most intense peak corresponds to the O-H bonds at 3360 cm^−1^, while the peak at 2930 cm^−1^ belongs to the strong bonds of alkanes (C-H). Similarly, the 1610 cm^−1^ peak shows the presence of alkene compounds (C=C), and the 1425 and 1060 cm^−1^ peaks confirm the presence of NO_2_ and C-H bonds [57,58]. As can be clearly seen, the intensity of the transmittance peaks decreases compared to the initial spectrum, which would be due to the degradation of microorganisms in the process of generating bioelectricity and sedimentation in the last days [59].

The regions sequenced and analyzed in the BLAST program obtained an identity percentage of 99.82%, which corresponds to the species *Wickerhamomyces anomalus* (see Table 1). This is a widespread yeast in nature, present in habitats such as soil, plants, and fruits and as an opportunistic pathogen in humans and animals [60]. This yeast can grow under extreme conditions of environmental stress, due to which it can be a spoilage organism, especially in food products with a high sugar content [61].

The dendrogram was based on the ITS regions of the rDNA regions of a group of yeast strains isolated from the anode plate of pineapple microbial fuel cells (Figure 5), which were constructed using the MEGA program, which relates and groups sequences of species; from this, the species *Wickerhamomyces anomalus* was identified, which is among the “film-forming” yeasts. In this context, it is worth mentioning that in microbial fuel cells, the transfer of electrons from microorganisms to the electrode is produced by various mechanisms, including through pili or nanowires after the formation of a biofilm on the electrode [62,63]. It has been reported that this species exhibits a direct transfer of electrons without the help of mediators, using glucose as a carbon source, and that this yeast also has redox enzymes present in the cell membrane of the cell, which contribute to the production of current in an MFC [64,65]. Finally, Figure 6 shows the bioelectricity generation mechanism of using pineapple waste as fuel in single-chamber microbial fuel cells, where it is observed that the three fuel cells connected in series were capable of generating 2.85 V, which was enough to turn on an LED bulb (white) on the fifteenth day.

## 4. Conclusive Remarks and Future Perspectives

Bioelectricity was successfully generated through single-chamber microbial fuel cells on a laboratory scale using pineapple waste and zinc and copper electrodes as a substrate. It was possible to observe that the maximum peaks of voltage and electric current were 0.99 ± 0.03 V and 4.95667 ± 0.54775 mA on the sixteenth and twentieth days, and these values decreased slowly until the end of the monitoring. These peak values of voltage and electrical current were obtained at an optimal pH of 5.21 ± 0.18, a substrate electrical conductivity of 145.16 ± 9.86 mS/cm, and two ° Brix. Likewise, the maximum power density found was 513.99 ± 6.54 mW/m^2^ with a current density of 6.123 A/m^2^, and the internal resistance of the microbial fuel cells was 865.845 ± 4.726 Ω, while the initial and final FTIR spectra of the substrate used were obtained, achieving a decrease in the transmittance peaks, the most notable being the peak belonging to the O-H bond at 3360 cm^−1^. Finally, the yeast *Wickerhamomyces anomalus* was molecularly identified as being present in the anode electrode with an identity percentage of 99.82%. This research highlights the importance of the use of pineapple waste in the generation of bioelectricity; with this substrate, electrical values higher than those found in the literature have been obtained. The molecular identification of the microorganisms present in the microbial fuel cells contributes to enriching the knowledge about the operation of microbial fuel cells. On the other hand, companies, society, and farmers can see an opportunity in which their waste can be reused, increasing their profits and benefits.

For future work, it is recommended that an optimal pH is used for the operation of the cells, since in this investigation, we worked with the pH of the pineapple waste itself; we further recommend using metal electrodes (due to their excellent electron-conducting properties) but coated with some type of non-toxic chemical compound so that microorganisms are not affected and thus increase the efficiency of microbial fuel cells. Carrying out these new investigations will clarify whether the metallic electrodes have a higher performance when they are coated or pure; this can be measured through the electrical values found in future investigations. On the other hand, the use of a cell system with air flows is recommended to increase power density, as it has been shown that the supply of O_2_ increases the electrical values, mainly the PD values [66], which, combined with sucrose-enriched substrates, can increase the voltage and current values [67].

## Figures and Tables

**Figure 4 molecules-27-07389-f004:**
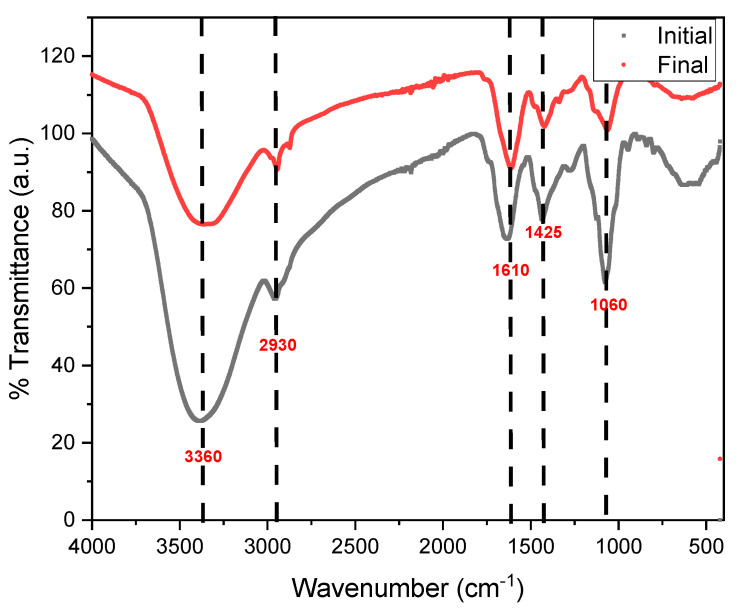
FTIR spectrophotometry of the initial and final pineapple waste.

**Figure 5 molecules-27-07389-f005:**
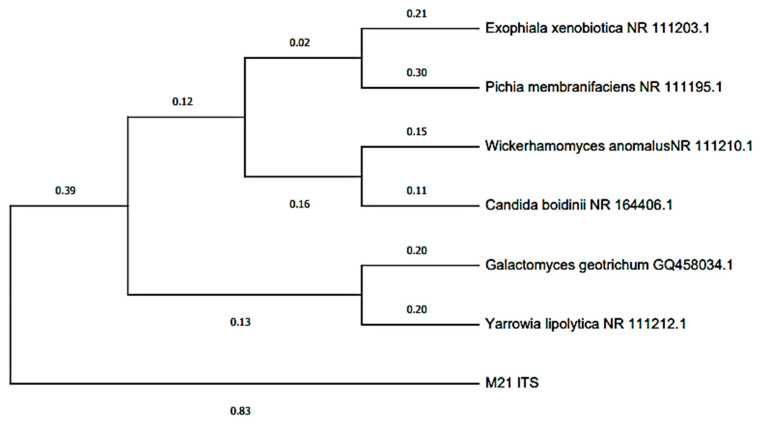
Dendrogram based on the ITS regions of the rDNA regions of a culture of *Wickerhamomyces anomalus* isolated from the MCC anode plate with pineapple substrate.

**Figure 6 molecules-27-07389-f006:**
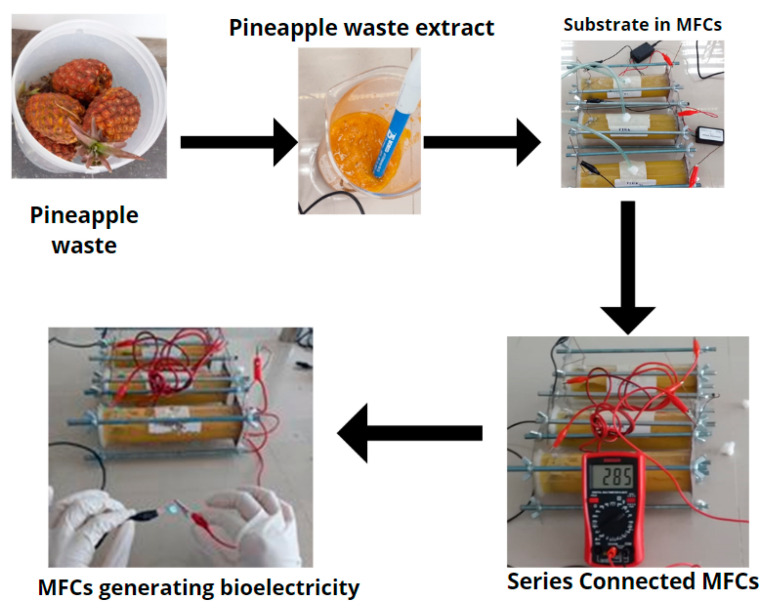
Mechanisms of the bioelectricity generation process from pineapple waste.

**Table 1 molecules-27-07389-t001:** BLAST characterization of the rDNA sequence of yeast isolated from the MFC anode plate with pineapple debris substrate.

Blast Characterization	Length of Consensus Sequence (nt)	% Maximum Identidad	Accession Number	Phylogeny
*Wickerhamomyces anomalus*	545	99.82	KJ527063.1	Cellular organisms; Eukaryota; Opisthokonta; Fungi; Dikarya; Ascomycota; Saccharomyceta; Saccharomycetales; Phaffomycetaceae; Wickerhamomyces

## Data Availability

Not applicable.

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
