# Peer review of "Use of Pineapple Waste as Fuel in Microbial Fuel Cell for the Generation of Bioelectricity"

_molecules, 2022, doi:10.3390/molecules27217389_

Round 1
Reviewer 1 Report
My comment to Author :
1- The page 2 line 88 it was found word Days what is that mean?
2- Page 2 line 91 from this Research will add till line 94 should be transfer to the conclusion
3- In Material and Methods
the statistical method should be added in page 3 with reference after title
2.5. Molecular Identification of Fungi under
2.6. Statistical analysis
Author Response
Dear Colleague, I hope you are doing well.
The authors welcome your comments and provide responses to each one.
I hope do you like it.
best regards
1.The page 2 line 88 it was found word Days what is that mean?
Ans. It was an error, it must be "32 days", it has been corrected
2. Page 2 line 91 from this Research will add till line 94 should be transfer to the conclusion
Ans. Changes were made
3- In Material and Methods
the statistical method should be added in page 3 with reference after title
2.5. Molecular Identification of Fungi under
2.6. Statistical analysis
Ans. Changes were made

Reviewer 2 Report
This manuscript dealt with electricity generation from pineapple waste by MFCs.
However, 1. there is no data about pineapple waste properties(composition or elemental analysis)
2. What is seed culture for MFCs?
3. How about present cellulose or hemicellulose change rather than FTIR data
4. Please modify Figure 3
5. In figure 4, Inicial -> Initial
6. Check the significant figures in the whole manuscript (ex 0.99 V 0.5685 V)
7. It contains over 60 references. Please reduce the ref.
8. In 2.1, unit 10 X 30 cm, not cm2
9. line 190, the pH varies from 4 to 5.5. this pH is quite a low pH environment for growing methanogen.
10. Author showed only yeast isolation data. Did you do bacteria isolation? And do you have NGS data?
Author Response
Dear Colleague, I hope you are doing well.
The authors welcome your comments and provide responses to each one.
I hope do you like it.
best regards
1. there is no data about pineapple waste properties(composition or elemental analysis)
Ans. The authors did not perform this initial analysis, but it will be taken into account for further research. Thanks for the point of view.
- What is seed culture for MFCs?
Ans. No type of initial culture was used, the waste was used as it is found in the environment, in order to simulate how it would be if it were applied in real life.
- How about present cellulose or hemicellulose change rather than FTIR data
Ans. FTIR was considered due to the compounds it would present, but it is a good characterization to perform it on cellulose or hemicellulose and it will be taken into account for a subsequent investigation.
- Please modify Figure 3
Ans. ok, it was fixed
- In figure 4, Inicial -> Initial
Ans. ok, it was fixed
- Check the significant figures in the whole manuscript (ex 0.99 V 0.5685 V)
Ans. have been reviewed and are correct.
- It contains over 60 references. Please reduce the ref.
Ans. The references are the ones used, if I reduce the references I would have to reduce the text, the authors, in order for the manuscript to have the greatest amount of information obtained from the reviews carried out, consider that the references should not be reduced.
- In 2.1, unit 10 X 30 cm, not cm2
Ans. ok, it was fixed
- line 190, the pH varies from 4 to 5.5. this pH is quite a low pH environment for growing methanogen.
Ans. pH values ​​shown are for substrates; The metagenomic culture was performed from the biofilm found on the anodic electrode and worked separately for its identification.
- Author showed only yeast isolation data. Did you do bacteria isolation? And do you have NGS data?
Ans. The isolation of bacteria and yeasts was carried out in selective media, only having growth in yeast medium, performing a staining to observe microscopic characteristics and then obtain a pure culture. From this axenic culture, molecular identification was performed by Sequence Analysis using the Sanger method.

Round 2
Reviewer 2 Report
What is seed culture for MFCs?Ans. No type of initial culture was used, the waste was used as it is found in the environment, in order to simulate how it would be if it were applied in real life.
- Hence, the microorganism of seed culture comes from pineapple and the authors' environment. It would be nice to check the microbial community over time because of mixed culture. In particular, there must be initial data.
Check the significant figures in the whole manuscript (ex 0.99 V 0.5685 V)
Ans. have been reviewed and are correct.
- please check again. ex) line 184 and 185.--> 189.3 and 199.33
Author showed only yeast isolation data. Did you do bacteria isolation? And do you have NGS data?
Ans. The isolation of bacteria and yeasts was carried out in selective media, only having growth in yeast medium, performing a staining to observe microscopic characteristics and then obtain a pure culture. From this axenic culture, molecular identification was performed by Sequence Analysis using the Sanger method.
- It is hard to believe this is pure culture because pineapple waste extract has enough carbohydrates to grow microorganisms.
Author Response
Dear colleague, I hope you are in good health.
I sent the answer to each comment, it should be noted that the authors carefully reviewed each point.
best regards
- What is seed culture for MFCs? Ans. No type of initial culture was used, the waste was used as it is found in the environment, in order to simulate how it would be if it were applied in real life.
- Hence, the microorganism of seed culture comes from pineapple and the authors' environment. It would be nice to check the microbial community over time because of mixed culture. In particular, there must be initial data.
Ans. On this occasion these initial tests were not carried out and only the final ones were considered; but it will be taken into account for further investigation.
2. Check the significant figures in the whole manuscript (ex 0.99 V 0.5685 V)
Ans. have been reviewed and are correct.
- please check again. ex) line 184 and 185.--> 189.3 and 199.33
Ans. Dear referee, these are the values ​​found for the electrical conductivity of the substrate used. The authors see no errors in the placed values.
3. Author showed only yeast isolation data. Did you do bacteria isolation? And do you have NGS data?
Ans. The isolation of bacteria and yeasts was carried out in selective media, only having growth in yeast medium, performing a staining to observe microscopic characteristics and then obtain a pure culture. From this axenic culture, molecular identification was performed by Sequence Analysis using the Sanger method.
- It is hard to believe this is pure culture because pineapple waste extract has enough carbohydrates to grow microorganisms.
Ans. This may be due to several factors, one of the most important according to the authors is the use of copper as an anodic electrode; to the elimination of many bacteria. Therefore, in future work it is recommended to cover the copper electrode with another non-toxic material. But that was the only microorganism found that he referred to.
